# Fire Behaviour Observation in Shrublands in Nova Scotia, Canada and Assessment of Aids to Operational Fire Behaviour Prediction

**Anne-Claude Pepin [1],* and Mike Wotton [2]**

[1] Parks Canada, Cape Breton Highlands National Park, 37486 Cabot Trail, Ingonish, NS B0C 1L0, Canada
[2] Canadian Forest Service, Graduate Department of Forestry, University of Toronto, Toronto, ON M5S 3B3, Canada; mike.wotton@utoronto.ca
* Correspondence: anne-claude.pepin@canada.ca

**Abstract:** Parks Canada, in collaboration with Nova Scotia Lands and Forests and Natural Resources Canada, documented shrub fire behaviour in multiple plots burned over two periods: a spring period in June 2014 and a summer period in July 2017. The study area, located within Cape Breton Highlands National Park, comprised fifteen burn units (20 m by 20 m in size). Each unit was ignited by line ignition and burned under a wide range of conditions. Pre-burn fuel characteristics were measured across the site and used to estimate pre-fire fuel load and post-fire fuel consumption. This fuel complex was similar to many flammable shrub types around the world, results show that this shrub fuel type had high elevated fuel loads ($3.17 \pm 0.84$ kg/m$^2$) composed of exposed live and dead stunted black spruce as well as ericaceous shrubs, mainly *Kalmia angustifolia* (evergreen) and *Rhodora canadensis* (deciduous). Data show that the dead moisture content in this fuel complex is systematically lower than expected from the traditional relationship between FFMC and moisture content in the Canadian Fire Weather Index System but was statistically correlated with Equilibrium Moisture Content. A significant inverse relationship between bulk density and fire rate of spread was observed as well as a clear seasonal effect between the spring burns and the summer burns, which is likely attributable to the increase in bulk density in the summer. Unlike most shrub research, wind and dead moisture content did not have a statistically significant association with fire spread rates. However, we believe this to be due to noise in wind data and small dataset. Rate of spread as high as 14 m/min and flame lengths over 4 m were recorded under Initial Spread Index values of 6.4 and relative humidity of 54%. A comparison with a number of well-known shrubland spread rate prediction models was made. An aid to operational fire prediction behaviour is proposed, using a fuel type from the Canadian Fire Prediction System (O-1b) and a modified estimate of fuel moisture of the elevated fuel in the fuel complex.

**Keywords:** shrub; fire behaviour; fuel moisture; fuel load; prescribed fire

---

## 1. Introduction

In Nova Scotia, shrublands, also known as coastal barrens or highland barrens, represent 6% of the province area and 12% of Cape Breton Highlands National Park (CBHNP) (Figure 1). Like elsewhere in the world, they are typically found in limiting edaphic and climatic growing conditions. In Nova Scotia, variations in composition and structure exist amongst various barrens, ranging from heath-dominated barrens to barrens with an increasing dominance of stunted spruce. At the Paquette Lake site, *Picea mariana*, *Kalmia angustifolia* and *Rhodora canadense* were the predominant species found.

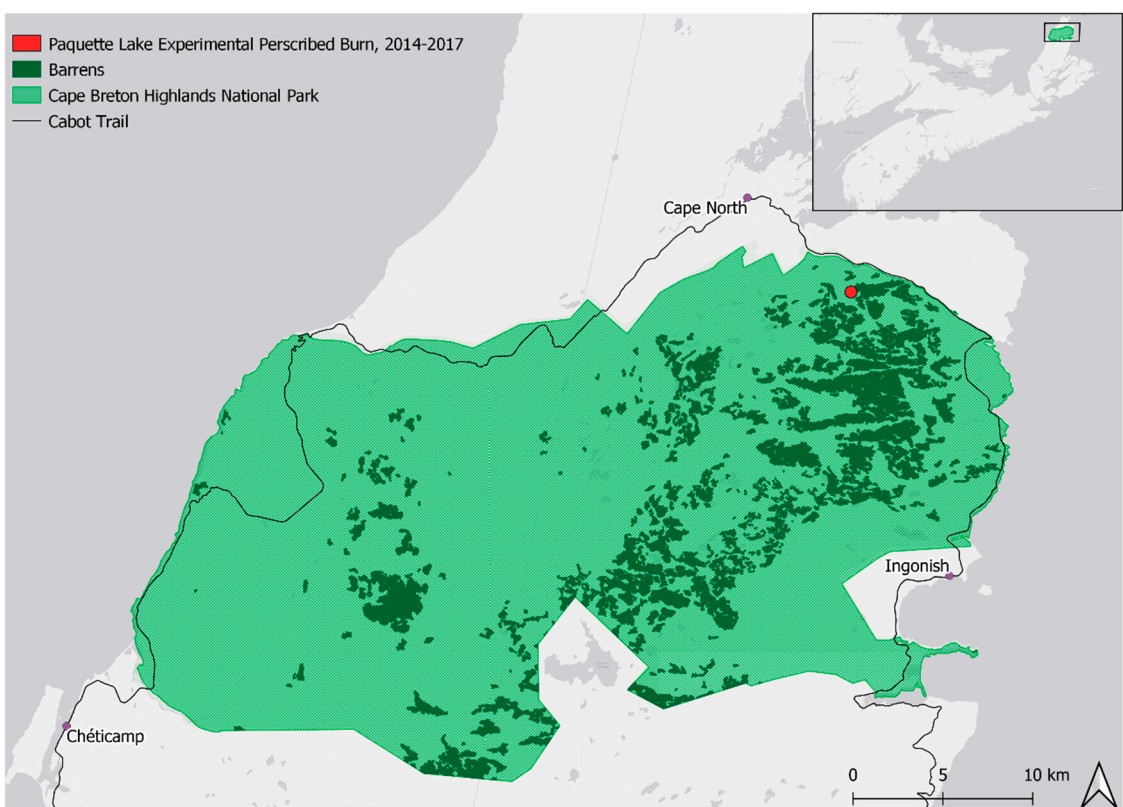

**Figure 1.** Extent of shrub barrens in Cape Breton Highlands National Park (CBHNP), Paquette Lake Experimental Prescribed Burn Site.

The very presence of barrens at latitudes and elevations where trees would be expected to grow is not well understood within CBHNP. Fire history data is also scarce for CBHNP but the barrens are thought to originate from severe fires dating back to the 1700s [1] although it appears other environmental factors contribute to maintaining them (e.g., shallow soils and wind [1]). Spruces half a meter high at the Paquette Lake Site were found to be around 60 years old.

Intense fire behaviour has been observed in this type of vegetation under weather conditions that would, in other more typical forest stands, be considered low risk. Operationally, these observed differences warranted a special fuel typing on the province's GIS fuel type layer; fire specialists in Canada refer to this fuel type as the "Nova Scotia Special Fuel Type (NS-1)". Although of different species composition, other shrub types in Australia, New Zealand, Portugal and Scotland exhibit extreme fire behaviour under low fire danger as well [2–6]. This has been commonly explained by the high proportion of dead fuel in the shrub canopy [7] and direct exposure to wind. Anderson et al. (2015) developed a shrub model pooling multiple datasets of existing shrub fire behaviour research from across the globe. Wind speed, fuel moisture and fuel height or bulk density were found to be the main factors driving fire behaviour [5].

The objective of this research project was to carry out and document a series of experimental fires in this novel NS-1 fuel type under a range of conditions and to assess the factors that influence fire behaviour, specifically spread rate. This latter objective was carried out both by examining the relationships between environmental factors (wind speed, moisture content and height or bulk density) and by comparing observed spread rates against prediction from existing models of shrubland fire behaviour from previous research. A further goal was to examine the ability to exist operational fire behaviour models (such as the U.S. BEHAVE System and the Canadian FBP System) to capture the observed variation in fire behaviour and hence be adapted easily into operational use in this new fuel type.

## 2. Materials and Methods

### 2.1. Burn Units and Fire Weather Station Establishment

The burn site is located in Northeast CBHNP along Paquette Lake road (46°50.2′ N, 60°25.9′ W) and comprises fifteen adjacent 20 m by 20 m burn units isolated from one another by a 4 m wide fire break (Figure 2).

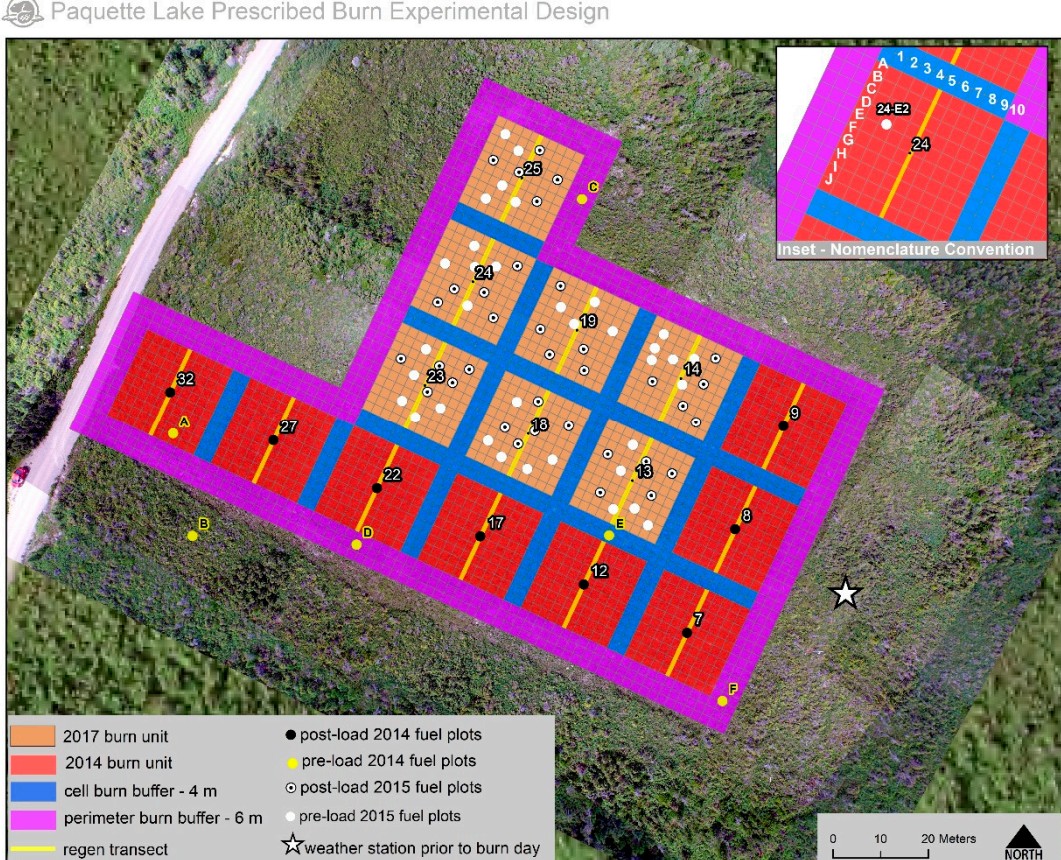

**Figure 2.** Burn unit layout and location of fuel loading sampling plots at the Paquette Lake Prescribed Fire Site in Cape Breton Highlands National Park.

The fire break was created by chipping the vegetation using an excavator with a mulching head. Additional fire control on burn days was provided using wet lining when necessary, taking extra precautions not to influence local relative humidity conditions within the plot being ignited.

The fifteen units were burned under a wide range of conditions to document fire behaviour and to help determine prescribed fire thresholds for future restoration burning activities. Eight of these units were burned between 10 June and 12 June 2014, prior to bud break (spring dormant season) while the remaining seven were burned under drier summer conditions and foliage out on the shrubs (summer growing season) on 14 to 15 July 2017. The burn units were all on relatively level ground, thus results should be relatively unaffected by the influence of slope.

One portable FTS weather station was set up within 20 m of the burn site one month prior to both burning periods. Observations of temperature, relative humidity, wind speed (height of 2 m) and precipitation were recorded hourly. Daily and hourly elements of the Canadian Fire Weather Index System [8] were calculated using 'Autocaller' software. Specifically, the traditional 1 pm fire weather observation was used to calculate the daily outputs for the FWI System [8] (including the daily Fine Fuel Moisture Code (FFMC)) and hourly weather was used to calculate an hourly version of the FFMC since fine fuel moisture can vary substantially throughout the day. We will hereafter

refer to this hourly weather-based calculation of the FFMC as the hFFMC, though in common use it is often also simply referred to as the FFMC or the hourly FFMC. The weather station was relocated during the burn operations to the closest open barren where it would not be at risk of burning, 770 m away. Although the station was set up to record wind speed every minute during the burning period, the one-minute interval data logging failed and therefore wind observations associated with each burn unit are taken from the top of the hour (running 10 min average) 770 m northwest from the site. However, for six of the fifteen burn units, live readings (running 10 min average) during the burn are recorded on videotape via a radio connection between the portable station (770 m away) and a handheld radio. These more precisely timed wind observations were used when available (Table A1). Wind speeds, recorded at 2 m, were multiplied by 1.31 to scale up to the standard 10 m observation height using the smooth wind adjustment factor for wind data collected between 2–2.9 m [8]. Each burn unit was ignited along the upwind edge of the unit starting at one corner going across the unit edge perpendicular to the dominant wind direction. Unit 18 had a diagonal ignition line at the south corner in order to have the ignition line perpendicular with the wind direction.

*2.2. Fuel Moisture*

Characterization of moisture content in various elements of the fuel complex followed the methodology developed by Oikle et al. in 2011 [9]. Three replicates of each fuel category type were sampled randomly from across the burn site on burn days. The various category types collected for moisture analysis were: shrub foliage, spruce needle foliage, live stems 0–1 cm; live stems > 1 cm; dead stems 0–1 cm; dead stems > 1 cm; litter and duff. Samples were sealed in containers, weighed before and after oven drying. The samples were dried for 24 h at 100 °C. This same procedure was used throughout the season over three years to examine the relationship between dead moisture content in the fuel complex and FWI System indices and to gain a better understanding of seasonal live moisture content fluctuations.

2.2.1. Live Moisture Content (LMC) Spring 2014

On 11 and 12 June 2014, fuel moisture samples were collected for foliage, live stems 0–1 cm and 1–7 cm in diameter (Table A2) (burn days were 10 and 12 June). The foliage was not out on deciduous shrub and *Rhodora canadense* was flowering and buds starting to flush. The foliage sample was a composite that was representative of the foliage types in the fuel complex (spruce needles, deciduous shrub, evergreen shrub, flowers and buds all mixed together).

2.2.2. LMC Summer 2017

On 16 July 2017, the day after burning was completed, live fuel moisture was again sampled from the fuel complex. The foliage was fully out on all shrubs and *Kalmia angustifolia* was flowering. In 2017 spruce needles were collected separately from the shrub foliage; the shrub foliage was a representative composite of deciduous, evergreen and flowers mixed together (Table A3).

2.2.3. Dead Moisture Content (MCdead)

Dead fine fuel moisture content is more variable day-to-day (and within the day) than live moisture as it responds quickly to changes in the weather. Samples of the dead and cured components of the fuel complex (0–1 cm diameter dead stemwood, >1 cm diameter dead stemwood, and litter) were collected on 12 different days during 2013, 2014 and 2015 to establish the relationship between actual dead moisture content and Canadian FWI System's weather-based moisture codes for this specific fuel complex (specifically the hourly Fine Fuel Moisture Code, hFFMC [10]). Early sampling had indicated no significant difference between moisture in the 0–0.5 cm and 0.5–1 cm diameter dead woody material and they were subsequently sampled together. Three additional replicates of each fuel component were collected at the time of the burning experiment, on 10 June 2014, 12 June 2014, and on 16 July 2017. For each sample time, in addition to the hFFMC estimate, the hourly weather data

collected at the site was also used to estimate the associated equilibrium moisture content (EMC) using the standard model used for fine fuel in the FWI System [8,10].

### 2.3. Vegetation Composition and Height

At the Paquette Lake site in 2014, vegetation height in each unit was not measured prior to the burn. Instead, visual assessments of vegetation height were estimated from the photographic documentation of the 2014 burn. In 2017, height was measured through vegetation transect and fuel loading quadrats. Vegetation transects were established in all of the burn units that were burned in 2017 following the point intercept method [11]. Every meter along the transect, a pin was lowered down and the height at which the tallest plant met the pin (visually average top of the canopy) was recorded. This method resulted in a total of 20 height observations per unit. One square meter quadrat per transect recorded per cent cover, stem density, average height and maximum height per species for each unit. In addition, five 1 m$^2$ fuel loading plots per unit were sampled in 2015. Per cent cover, average stem height and maximum stem height were recorded per species for each of the 7 units. For those loading plots, the average height per unit was calculated by averaging all shrub canopy species' average height, while leaving out low ground species such as bog laurel (*Kalmia polifolia*), bunchberry (*Cornus Canadensis*), dead woody debris, goldthread (*Coptis trifolia*), American wintergreen (*Pyrola Americana*), tea berry (*Gaultheria procumbens*), lichens and mosses. The final height associated with each burn unit was calculated by using an average of the two methods (vegetation transects and fuel loading quadrats); these measured heights were consistent with ocular estimates of heights from photos.

### 2.4. Fuel Load and Fuel Consumption

Prior to burning in 2014, fuel load was characterized across the entire study area using six one squared metre sample plots in total. These sample plots were not specifically located within each burn unit (Figure 2), therefore, in subsequent analysis fuel load was assumed to be the same across the site for the 2014 burn units. Within those plots, all material was collected per sample type, including foliage, live stems 0–1 cm, live stems > 1 cm, dead stems 0–1 cm, dead stems > 1 cm, litter and duff. Samples were dried in an oven and the dry biomass weight. In 2015, five 1 m$^2$ fuel load plots were sampled on each of the seven remaining unburned plots (n = 35 overall) following the same methodology but increasing the sampling effort to obtain burn unit-specific fuel loading results. It took until 2017 to be in prescription and conduct the burn operation.

Burn unit-specific post fuel loads were collected for both 2014 (n = 3/burn unit) and 2017 burn units (n = 5/burn unit) using the same sampling method. For the 2014 burns fuel consumption was calculated using the overall pre-burn fuel load average value minus the burn unit-specific post load results. For the 2017 burns consumption was estimated using the difference between plot specific pre-burn and post-burn fuel load assessments.

### 2.5. Fire Behaviour

#### 2.5.1. Rate of Spread (ROS)

Each burn unit was gridded with twenty-five 3 m tall rebar posts that were 5 m apart (Figure A1). Human observers located on each side of the ignition line recorded time at which the fire front reached each stake. To infer a rate of spread for each burn unit based on these observations, a start and end stake for the frontal rate of spread are identified based on wind direction, ignition line position, pictures and videos; the starting stake for this calculation was chosen considering that the ignition line needed some time to become established. The rate of spread (m/min) is then calculated using the following formula:

$$\text{ROS} = \text{Distance between the start and end stakes}/(\text{Time at End stake} - \text{Time at Start stake}) \quad (1)$$

In 2017, the timing of fire arrival throughout the grid was also estimated from the output of thermocouples placed at selected grid points in each unit. These were type K thermocouples with output recorded at 1 s intervals. Each thermocouple was placed at a height of 30 cm in vegetation and captured the arrival time of the fire front at that location. Arrival time of the front was determined to be when the temperature reached 300 °C and exceeded that threshold for at least three consecutive seconds. Arrival times at points were used with Simard et al.'s triangulation method [12] to estimate spread rate and direction. Only those sets of timers where calculated direction matched the documented headfire direction were used as estimates of spread rate; thermocouple arrival times at stake locations were excluded if either calculated spread direction or observer notes implied flank fire spread was occurring through that location.

To validate the observer-based method of ROS estimation used in 2014, this same method was also used with 2017 burn unit observations and compared to the spread rate estimates obtained with the thermocouple-based electronic timers; both methods gave similar estimates with no noticeable bias.

### 2.5.2. ROS Model Comparison

A number of models exist in the literature that describes the spread rate in various shrubland fuel complexes. With observations possible over a total of only 15 burn units, and also split between spring and summer conditions, we expected there would not be enough data spanning a broad enough range of conditions to develop a completely new robust model; therefore, the ability of several of these existing models [2–5] to predict spread during our burns was explored. For Anderson et al.'s recent general shrubland model [5] two models are presented base on their crown bulk density and shrub height formulation, and their adjustment for line length is also applied to the data.

In this comparison, we included a comparison with the US and Canadian operational fire behaviour prediction system. For the US BEHAVE model [13] we used the SH-6 model, which we believe best described this fuel type. For the Canadian FBP System [14], which does not have a shrubland type at all, we chose one of the open fuel types, O-1b. While O-1b represents a large open fuel complex of standing grass, we felt that it likely captured the most appropriate interaction between open wind (unattenuated by forest canopy) and flame within this fuel complex. The FBP System models use daily or hourly FFMC (an indicator of litter moisture on the forest floor of a closed pine canopy stand) and wind to calculate the Initial Spread Index (ISI) [10,14] which in turn drives spread rate predictions. As it is known that the hFFMC does not track moisture in open exposed fuels consistently, we also tested a modified version of the O-1b model which is not formally part of the FBP System [14] but has been used in a similar way, with good predictive success, in other models of spread rate in open, exposed fine fuels [15]. Moisture content in the fast-drying, elevated fine fuels of the NS-1 fuel complex is estimated from the FWI System's estimate of Equilibrium Moisture Content (EMC) instead of from the hFFMC. This moisture content is then used to estimate ISI using the standard methods [14] and finally a spread rate using the existing FBP System relationship between ISI and spread rate for O-1b (we hereafter call this, 'O-1b modified').

### 2.5.3. Fire Intensity and Flame Length (m) Relationships Models

Average flame length of the fire front is estimated from pictures and videos using the 3 m rebar stakes throughout the burn units as reference points (colour alternated every 50 cm along the length of the rebar). Flame lengths were estimated by two people independently and then the final length was agreed upon between the two observers.

Fire intensity was estimated both from observed estimates of flame length and using Byram's classic equation for fireline intensity [16]:

$$I_B = H \times W \times R \text{ kW/m} \qquad (2)$$

In Equation (2) H is the low heat of combustion for the fuel (as described below) (kJ/kg), W is the fuel load consumed (kg/m$^2$) and R is the observed spread rate (m/s). There are numerous flame lengths to fireline intensity relationships in the literature [17], For this study, we examined the relationship between observed flame lengths and calculated fireline intensity (from Byram's equation [Equation (2)]) and contrasted these data with several of these existing flame length-intensity models from the literature [3,17–20]. We fitted a simple power function to our data to characterize this flame length vs intensity relationship for the NS-1 fuel type, similar to the form used by others.

### 2.5.4. Heat of Combustion

Since the NS-1 fuel type was potentially different than other standard fuels in Canada, the heat of combustion for elements of the NS-1 fuel complex was estimated in the lab with a Parr Instruments 1341 Plain Jacket oxygen bomb calorimeter. Samples of the fuel complex were ground and passed through a 2 mm mesh sieve; 0.5–0.7g of this sample was combusted within the calorimeter. The heat of combustion was measured separately for the stems and foliage of the two main shrub species (*Kalmia angustifolia* and *Rhodora canadense*). For the other main fire carrying component of the fuel complex, *Picea mariana*, a well-established heat of combustion was used [21].

## 3. Results

Local fuel moisture codes and fire behaviour indices of the FWI System at the time of burning and corresponding observed fire behaviour are shown in Table 1. The ROS value for each experimental fire is an average over the size of the burn unit for the head fire and ranged between 3.1 m/min and 13.9 m/min. Headfire fireline intensity values ranged between 1475 kW/m and 8750 kW/m.

### 3.1. Fuel Load, Vegetation Height and Bulk Density

Average coverage across the study site was: *Picea mariana* (30% cover live (s.d. 16) and 5% cover dead (s.d. 7)), *Kalmia angustifolia* (19% cover (s.d. 13); a flowering shrub with evergreen leaves also of the Ericaceous family) and *Rodhora canadiense* (14% cover (s.d. 7); a flowering shrub with deciduous foliage of the Ericaceous family) along with some *Cladonia*, *Pleurozium schreberi* (feather moss) and *Sphagnum* spp. underneath. The average percent cover across the site for less predominant species was *Illex micrunata* (5% (s.d. 5)), *Vaccinum angustifolium* (7% (s.d. 6)), *Kalmia latifolia* (1% s.d. 1), *Amelancia* sp. (1%, (s.d. 3)), *Viburnum nudum* (1%, (s.d. 12)), *Pteridium aquilinum* (1% s.d. 1) and *Cornus canadensis* (1% s.d. 1).

For 2014, the average litter load was 0.8 kg/m$^2$ (s.d. 0.14) and the duff layer load was 10.6 kg/m$^2$ (s.d. 3.2). Above ground (elevated, with litter) fuel loading was 3.3 kg/m$^2$ (s.d. 1.2) or 2.7 kg/m$^2$ (s.d. 1.1) without litter which, with the average fuel complex height of 0.5 m, corresponded to an above ground bulk density for the fuel complex of 5.3 kg/m$^3$ (s.d. 1.8).

For 2017, the average above-ground fuel load prior to the burn was 3.2 kg/m$^2$ (s.d. 0.8) with litter load making up 1.1 kg/m$^2$ (s.d. 0.4) of this total corresponding to an average litter depth of 3.0 cm (s.d. 1.4). The fuel complex overall had an average height of 0.48 m (s.d. 0.18). The average bulk density of the shrub layer (not including litter) was 4.4 kg/m$^3$ (s.d. 0.9). There were hummock formations within the fuel complex and therefore the duff layer depth was quite variable. The average duff load of 13.8 kg/ m$^2$ (s.d. 5.8) and a corresponding average duff depth of 19 cm (s.d. 14).

**Table 1.** Canadian Forest Weather Index (FWI) System component values associated with fifteen 0.04 ha experimental fires (2014–2017) along with corresponding fuel and fire behaviour observations.

| Unit | Date | Temp (○C) | RH (%) | 10 m WS (kph) | HFFMC | HISI | BUI | Fuel Load (kg/m²) | Vegetation Height (m) | Bulk Density of Shrub Layer * (kg/m³) | Frontal ROS (m/min) | Average Frontal Flame Length (m) | Average Frontal Intensity Flame Length ** (kw/m) | Fuel Consumption (kg/m²) | Frontal Intensity Byram Equation (kW/m) | EMC *** (%) |
|---|---|---|---|---|---|---|---|---|---|---|---|---|---|---|---|---|
| 7 | 06/10/2014 15:01 | 19.4 | 50 | 16 | 88 | 7.4 | 23 | 3.34 ± 1.1 | 0.5 | 5 | 8.4 | 4.1 | 5583 | 2.3 | 6025 | 13.8 |
| 12 | 06/10/2014 15:45 | 18.4 | 54 | 13 | 88 | 6.4 | 23 | 3.34 ± 1.1 | 0.5 | 5 | 13.9 | 4.3 | 6192 | 2.1 | 8750 | 14.7 |
| 17 | 06/10/2014 16:27 | 15.9 | 59 | 13 | 88 | 6.1 | 23 | 3.34 ± 1.1 | 0.5 | 5 | 6.9 | 4.4 | 6510 | 3.2 | 6949 | 16.1 |
| 8 | 06/10/2014 18:57 | 12.3 | 72 | 11 | 88 | 5.2 | 23 | 3.34 ± 1.1 | 0.5 | 5 | 5.7 | 3.7 | 4466 | 2.4 | 4311 | 19.4 |
| 9 | 06/10/2014 19:35 | 10.8 | 80 | 9 | 87 | 3.7 | 23 | 3.34 ± 1.1 | 0.5 | 5 | 4.4 | 3.3 | 3483 | 3.1 | 4226 | 21.8 |
| 32 | 06/12/2014 11:27 | 12.4 | 71 | 12 | 79 | 1.9 | 25 | 3.34 ± 1.1 | 0.5 | 5 | 3.5 | 3.1 | 3040 | 1.9 | 2099 | 19.2 |
| 27 **** | 06/12/2014 13:28 | 14 | 63 | 9 | 80 | 1.8 | 27 | 3.34 ± 1.1 | 0.5 | 5 | 2.0 | 2.8 | 2437 | 3.0 | 1880 | 17.2 |
| 22 | 06/12/2014 15:14 | 14.7 | 55 | 8 | 81 | 1.9 | 27 | 3.34 ± 1.1 | 0.5 | 5 | 3.4 | 3.3 | 3483 | 3.1 | 3288 | 15.6 |
| 13 | 07/14/2017 10:58 | 22.2 | 35 | 14 | 88 | 7.1 | 51 | 3.15 ± 1.1 | 0.58 | 4.46 | 4.9 | 4.0 | 5291 | 3.1 | 4782 | 10.4 |
| 14 | 07/14/2017 13:21 | 23.5 | 38 | 14 | 89 | 7.7 | 51 | 2.09 ± 0.6 | 0.35 | 3.41 | 7.1 | 5.0 | 8595 | 1.5 | 3313 | 10.8 |
| 24 | 07/14/2017 15:13 | 20.1 | 51 | 20 | 89 | 11 | 51 | 4.86 ± 3.6 | 0.58 | 3.52 | 5.1 | 6.0 | 12,776 | 3.0 | 4853 | 13.6 |
| 25 | 07/14/2017 17:08 | 20 | 57 | 15 | 89 | 8.6 | 51 | 2.77 ± 0.6 | 0.44 | 4.59 | 4.1 | 4.0 | 5291 | 2.5 | 3251 | 13.7 |
| 19 | 07/14/2017 18:05 | 18.2 | 63 | 16 | 89 | 8.6 | 51 | 3.12 ± 1.9 | 0.46 | 5.16 | 3.1 | 3.0 | 2831 | 1.5 | 1475 | 16.5 |
| 23 **** | 07/15/2017 10:52 | 23.2 | 37 | 20 | 89 | 11.1 | 51 | 3.19 ± 1.5 | 0.48 | 5.84 | 1.5 | 2.0 | 1173 | 2.1 | 999 | 10.6 |
| 18 | 07/15/2017 12:15 | 22.7 | 36 | 20 | 89 | 13.8 | 51 | 3.05 ± 1.1 | 0.46 | 3.89 | 5.0 | 4.5 | 6836 | 1.9 | 3013 | 10.6 |

* Bulk density of shrub layer excludes litter load; ** Intensity estimated from flame length and Byram's flame length relationship [12]. *** Dead moisture content of stems 0–1 cm was collected on 12 June 2014 at 1700 = 11.90%; and on 11 June 2014 at 11:00 = 13.65% and on 16 July 2017 between 1600 and 1800; =13.38%; **** Unit 27 and 23 were discarded due to many wind shift during the burn period. Unit 27 had two ignition line due to wind shift after 1st ignition.

Figure 3 shows the proportion of each sample type per burn unit. Actual averages per burn unit are presented in Table A4. Overall, the relative composition of the fuel complex appears similar in each burn unit except that unit 24 had a higher litter fuel loading than the other units and unit 14 had a lower overall load than the other units.

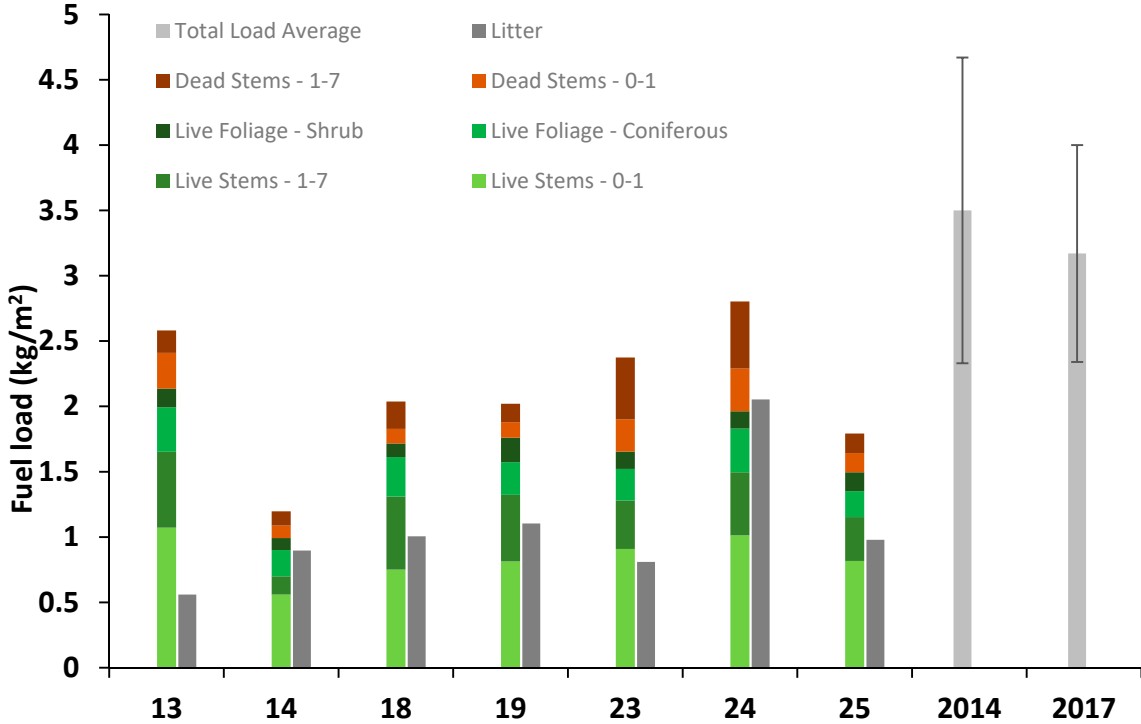

**Figure 3.** Fuel loading distribution by fuel complex component for study site 2014 and the more detailed plot specific sampling of each of the 2017 burn units.

### 3.2. Fuel Moisture

#### 3.2.1. Live Moisture Content (LMC), Spring 2014

The live foliar moisture content average was 169% (standard error (s.e.) 10%) on 11 June and 225% (only one sample, replicates were lost) on 12 June. Overall the foliar LMC was 183% (s.e. 16) when grouping those four samples together. Overall, from a fire perspective, foliar moisture was quite high, but these levels of moisture are consistent with the emerging deciduous vegetation on these dates (see Figure 4a). Annuals had not yet fully emerged, however, *Rhododendron canadense* (deciduous shrub) was in full bloom.

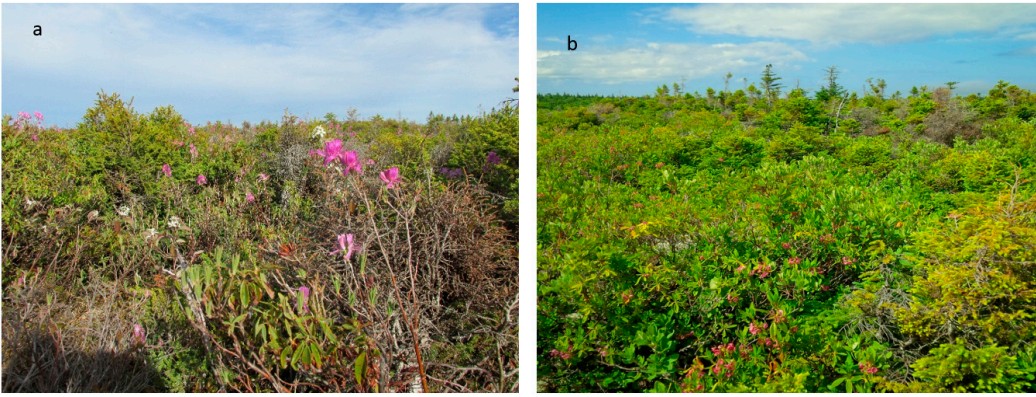

**Figure 4.** Picture of fuel Nova Scotia Special Fuel Type (NS-1) on (**a**) 12 June 2014 (**b**) 15 July 2017.

The average live moisture content of stems in the 0–1 cm diameter class was 101% (s.e. 3%). Since only 10% of the live stems in the 1–7 cm diameter class were observed to consumed in the 2017 burns, stems > 1cm category were left out of the LMC estimate. The fuel consumption in this class is assumed to have been less during the 2014 burns under lower associated Build Up Index (BUI) values from the FWI System (BUI is a numeric rating of the total of fuel available for combustion). The fuel loading between spruce foliage (without the broadleaf component) and the 0–1 cm diameter live stems from the 35 plots sampled in 2015 was 0.3 and 0.8 kg/m$^2$ respectively. Using these values, overall weighted moisture content was estimated as 122% for the fuel complex during the 2014 spring burns (Table A2).

### 3.2.2. Live Moisture Content (LMC), Summer 2017

Moisture contents of the live component fuels were: spruce foliage, 155% (s.e. 14%); shrub foliage, 207% (s.e. 50%); and, the 0–1 cm diameter stems, 105% (s.e. 4%). Since stem consumption was observed to be limited to the 0–1 cm category, stems > 1 cm diameter class were not considered. Using the relative fuel loading between shrub foliage, spruce foliage and 0–1 cm diameter live stems (0.1, 0.3 and 0.8 kg/m$^2$ respectively) overall weighted moisture content was estimated as 127% for the fuel complex during the 2017 summer burns (Table A3). Unfortunately, during sampling deciduous and evergreen shrub material was mixed together but likely have very different moisture contents, which can explain the high variation within this sample type.

### 3.2.3. Dead Moisture Content (MC$_{dead}$)

Figure 5a shows the moisture content found in dead stems between 0 and 1 cm was considerably lower than what would be expected from the standard FF-scale calibration. This was generally the case for dead stems > 1 cm as well. This woody material was dead and cured stemwood elevated within the fuel complex, and thus well-separated from the moisture within the forest floor. The hFFMC has been designed to track moisture of fine cured material sitting on and influenced by moisture within, a typical organic layer in the boreal pine forest. The elevated components of the NS-1 fuel complex are more exposed to the drying influences of wind and sun and separated physically from moisture in the fuel complex organic material. Such fast-reacting components in Canadian fuel types are sometimes estimated on the daily time scale by using the equilibrium moisture content (EMC) from the FWI System [22,23]. Figure 5b shows the relationship between observed moisture content in the 0–1 cm diameters fuel class and EMC from the FWI System [10].

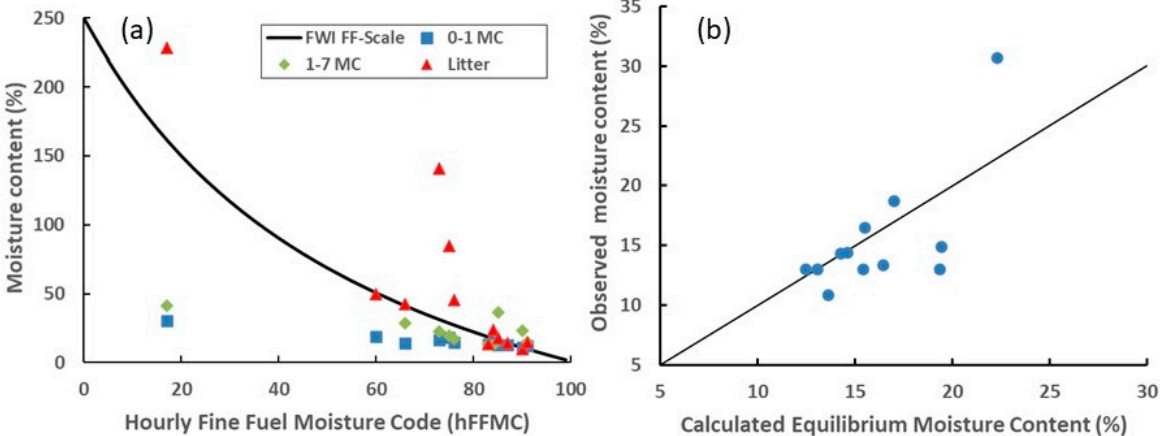

**Figure 5.** (**a**) Observed moisture content plotted against calculated hourly Fine Fuel Moisture Code (hFFMC) value for sample time for each fine fuel class sampled (overall n = 33) within in Nova Scotia Special fuel type (the line represented the standard FWI System 'FF-scale' model used for transforming fine fuel moisture content to its FWI System 'Code' value [8]). (**b**) the relation between litter Equilibrium Moisture Content (%) ([10]) and observed elevated dead moisture in the 0–1 cm diameter class (r = 0.7; $p$ = 0.01); the line of identity (y = x) is shown.

The litter is composed of reindeer lichen (*Cladonia* sp.), feather moss and *Sphagnum* moss along with some shrub's dead leaves. Across the full range of hFFMC observed, litter moisture followed the general trend suggested by the FWI System's FF-scale calibration between forest floor litter and FFMC (Figure 5a). However, there were some sample days where there was considerable variation and because of the forest floor composition, it is possible that certain samples had more *Sphagnum* in them and thus caused some stronger deviation (towards wetter litter) than the standard conifer litter calibration, which has been shown to work reasonably in a number of litter types [23].

### 3.3. Low Heat of Combustion

The low heat of combustion was measured for the main canopy species found in the fuel complex. Results were not significantly different from the FBP System default value of 18 MJ (Table 2).

**Table 2.** Low heat of combustion measured for various components of the NS-1 fuel complex.

| Fuel Component | Average Low Heat of Combustion (100% MC) |
|---|---|
| Kalmia new foliage | 19.03 (s.d. 0.46) |
| Kalmia winter | 20.23 (s.d. 2.26) |
| Rhodara | 19.45 (s.d. 1.34) |
| Stems mixed | 19.39 (s.d. 2.68) |
| Duff | 15.86 (s.d. 0.81) |

### 3.4. Fire Behaviour

### 3.4.1. Fuel Consumption

Overall consumption of fine fuels throughout the plots were fairly complete. All surface litter and foliage were completely consumed as was most of the 0–1 cm dead fuel load. Postburn measurement showed 0.2 kg/m$^2$ (s.d. 0.1) left in plots, and a total fuel consumption (litter, foliage and stem 0–1 cm) of 3.0 kg/m$^2$ (Table A4). Material in 1–7 cm diameter size class mostly remained after the passage of the flaming front, and while it was measured in the resampling, variability was so high that no significant difference from pre-burn load amount in this category could be found. These observations are consistent with observations of branchwood consumption in crown fires as well [24].

### 3.4.2. Fire Spread Rate

Previous modelling has identified wind speed, dead fuel moisture and bulk density as having an influence on spread rate in shrub fuel types. The relation between our observed spread rates and these elements of the fire environment are shown in Figure 6. Neither the springtime burns (2014) nor the summer burns (2017) showed the expected strong correlation with wind (in spring r = 0.6, $p$ = 0.2; in summer r = 0.1, $p$ = 0.8) (Figure 6a).

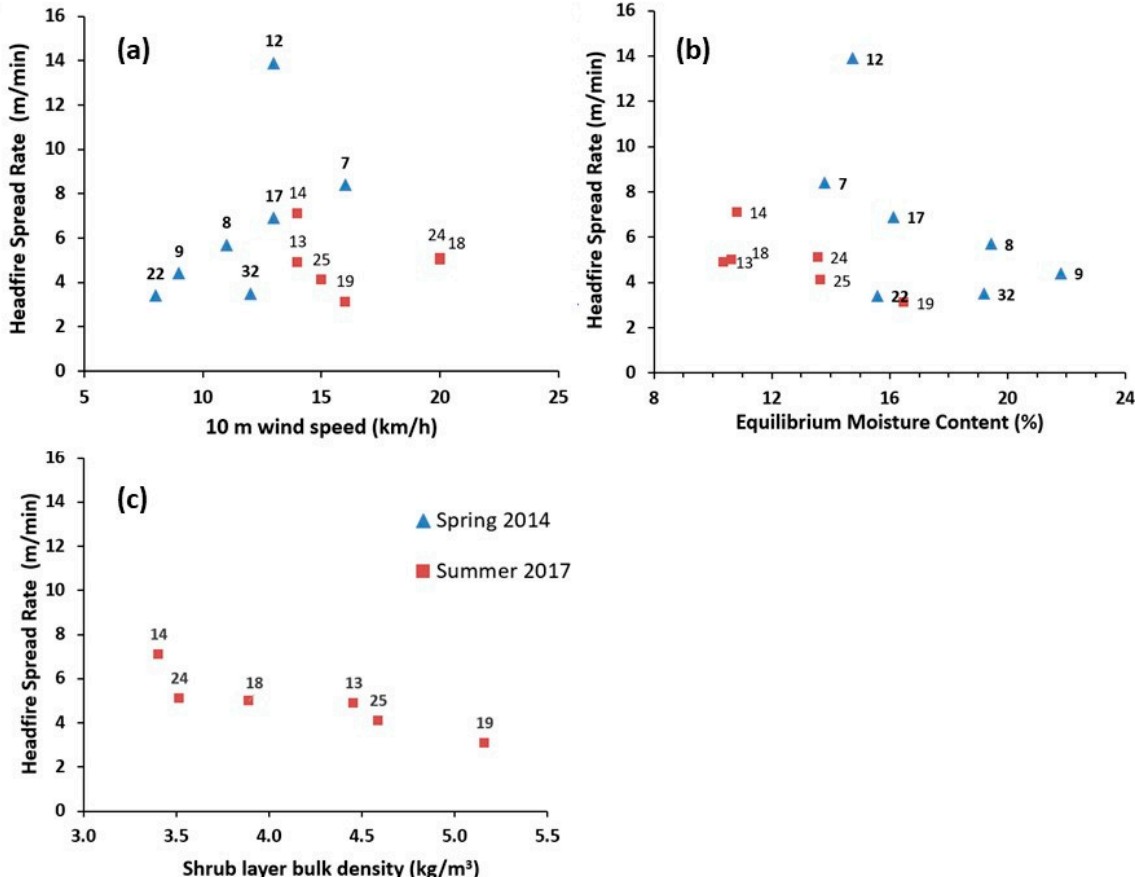

**Figure 6.** Observed spread rates compared with recognized factors previous studies have identified as influencing fire behaviour. (**a**) 10 m open wind speed (km/h), (**b**) dead fine fuel moisture content in elevated fuels (%) and (**c**) shrub layer bulk density (kg/m³).

Dead fuel moisture (estimated using the EMC model described in Figure 5b) is shown in Figure 6b and indicated the expected negative correlation (although not significant) with spread rate within each season, though across spring and summer the relationship appeared to be shifted, differentiating the two seasons (spring: r = 0.6; $p$ = 0.2; summer: r = 0.7; $p$ = 0.1).

For the burns where we had plot specific bulk density estimates (summer burns in 2017), a statistically significant negative relationship was found between observed spread rate and shrub layer bulk density (r = 0.9, $p$ = 0.02) (Figure 6c). Unfortunately, our first set of burns (2014) did not have this unit-to-unit differentiation in bulk density data.

### 3.4.3. ROS Model Comparison

A comparison with existing shrub fuel complex spread rate models in the literature is presented in Figure 7. Model prediction accuracy and references for each model are summarized in Table 3. None of the models reveals strong predictive performance, especially for the summer dataset. The modified O-1b FBP fuel type (which uses EMC instead of hFFMC), and the Anderson et al. models [5] give

reasonable results in the spring (r = 0.64, *p* = 0.12 and r = 0.67, *p* = 0.1 respectively). The original (unmodified) O-1b model in FBP has a reasonable correlation in spring, but a very high bias (Table 3).

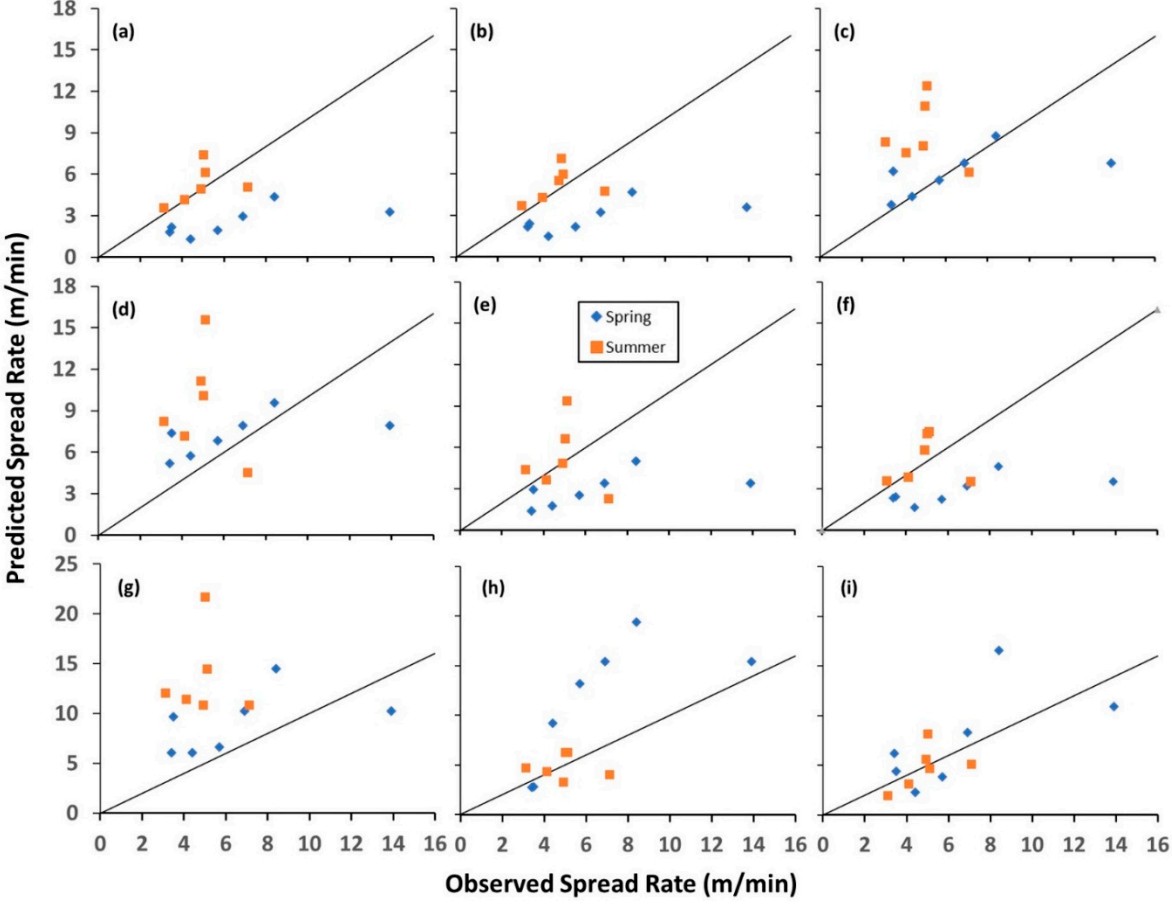

**Figure 7.** Rate of spread (m/min) for Paquette lake burning conditions using existing shrubland fire behaviour models: (**a**) Anderson et al.'s 2015 height model [5], (**b**) Anderson et al.'s 2015 bulk density model [5], (**c**) Catchpole et al. 1998 [18], (**d**) Davies et al. 2009 [2], (**e**) Fernandes 2000 [3], (**f**) Fernandes 2001 [4], (**g**) US Behave (SH-6) [13]; (**h**) FBP (O-1b) [14]; (**i**) FBP (O-1b modified) similar to [15].

**Table 3.** Comparison of results from Paquette lake with existing shrubland fire spread rate models.

| Model | Spring (n = 7) | | | Summer (n = 6) | | |
|---|---|---|---|---|---|---|
| | r (p) * | Bias * | MAE * | r (p) | Bias | MAE |
| Anderson et al. (2015) [5]: bulk density | 0.67 (0.10) | 4.1 | 4.2 | 0.42 (0.4) | −1.0 | 1.6 |
| Anderson et al. (2015) [5]: height | 0.67 (0.10) | 3.8 | 3.8 | 0.34 (0.5) | −0.5 | 1.2 |
| Catchpole et al. (1998) [18] | 0.58 (0.17) | 0.54 | 3.1 | −0.2 (0.7) | 4.0 | 4.4 |
| Davies (2009) [2] | 0.58 (0.17) | −0.63 | 2.4 | 0.22 (0.9) | 3.8 | 3.9 |
| Fernandes et al. (2000) [3]: height | 0.57 (0.18) | 3.7 | 3.7 | −0.17 (0.75) | 0.34 | 2.1 |
| Fernandes (2001) [4] | 0.64 (0.12) | 3.8 | 2.9 | −0.07 (0.88) | 0.12 | 1.4 |
| US BEHAVE [13] | 0.57 (0.23) | −2.4 | 3.5 | −0.02 (0.97) | 8.7 | 8.7 |
| FBP (O-1b) [14] | 0.71 (0.07) | −4.6 | 5.0 | −0.08 (0.87) | −0.02 | 1.5 |
| FBP (O-1b-modified) [14,15] ** | 0.64 (0.12) | −0.9 | 2.9 | 0.52 (0.28) | −0.09 | 1.4 |

\* r (p) is the correlation coefficient and its associated statistical significance *p*-value. Bias is average of Predicted–Observed. MAE is mean absolute error. \*\* Modification uses the EMC model (from the FFMC model in the FWI System [10]) to make an estimate of moisture to replace that used from FFMC in the ISI calculation (see Appendix A Table A7).

The summertime O-1b model is calculated using 60% of the fuel complex as being cured (i.e., 40% green), springtime calculation uses 100%. This curing effect on spread rate was as per the FBP System [14,25].

### 3.4.4. Fire Intensity and Flame Length

Numerous researchers have correlated Byram's fireline intensity to flame length for a range of specific fuel types [17]. We examined the relationship between observed flame lengths and calculated fire intensity (from Byram's equation [Equation (2)]) and contrasted these data with several of these existing flame length -intensity models from the literature (Figure 8).

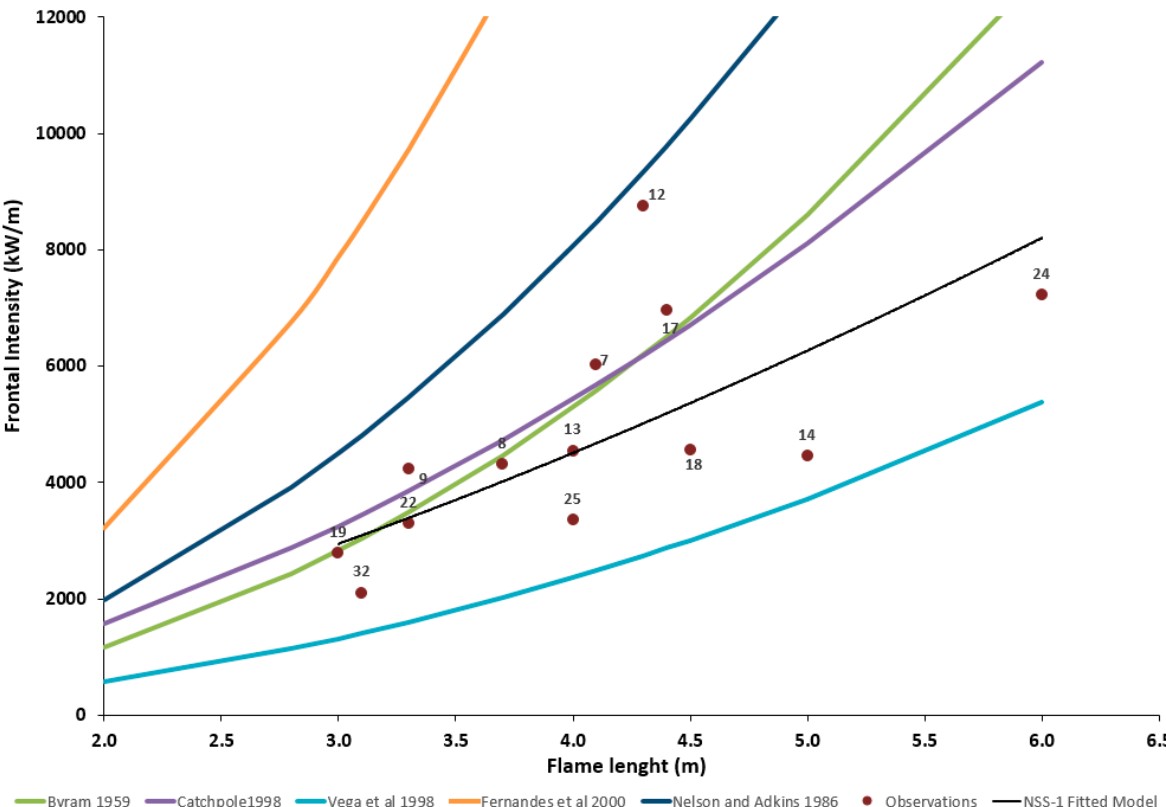

**Figure 8.** Observed flame lengths (m) and fireline intensity (in kW/m as calculated by Byram's equation (Equation (2)) along with several existing relationships from other studies.

Our data showed trends similar to other relationships in the literature, though did not track any existing relationship in particular. We fit a simple power function to our data to characterize this flame length vs intensity relationship for the NS-1 fuel type similar to the form used by others.

$$\text{FLI} = 581 \times \text{FL}^{1.5} \ (r^2 = 0.53, \text{n} = 13) \ \text{kW/m} \ (\text{FL} = \text{Flame length in m}) \tag{3}$$

## 4. Discussion

Wind speed is regularly observed to be the primary driver of the observed variability in fire spread rates in shrubland fire behaviour studies [3–5]. The wind is the primary factor in active crown fire spread rate as well [26]. At Paquette Lake, the wind did clearly seem to influence spread rate as expected. However, in our subsequent analysis, no statistically significant correlation was found between wind speed and spread rate in either the spring or summer dataset (Figure 6a). Wind speed on each fire was only available using 10-min average observations from the top of the previous or subsequent hour at the weather station, which was located 700 m away during the burning operations.

This wind data, as well as some gust observations, are presented in Table A1. These data reveal considerable variability in wind speeds from hour-to-hour and some significant gusting present. This variability could explain outliers such as burn unit 12, a burn where the headfire passage through the burn unit lasted only about one minute and could have been influenced by gusting observed during the hour around the burn itself. If the peak gust measured, of 31 km/h, were used, the correlation for the spring burn between ROS and wind speed would have been r = 0.9 ($p = 0.02$) instead of r = 0.6 ($p = 0.2$) when we use the average top of hourly wind speed of 13 km/h. The summer relationship between ROS and wind was noisier. The shifts in wind direction during the summer burn operation made it challenging, at times, to determine which side of the unit to ignite and resulted in spread rate estimates from one burn unit being discarded due to significant changes in the direction of the head fire over the short duration of the burn.

The increased bulk density due to the fully flushed fuel complex in the summer burns may also have contributed to our inability to see any wind-spread rate association in this summer period. Unfortunately, we did not have burn unit-specific measurements of bulk density for our set of spring burns, which took place before annual lead flush had finished, and we cannot examine this trend further.

Fire intensities and spread rates observed in the summer burns were similar to those observed in spring, although summer burns generally were conducted under higher wind speed, lower relative humidity, lower dead fuel moisture content and higher BUI conditions (Table 1). This suggests a rate of spread inhibiting factor in summer conditions; the presence of annual vegetation in the elevated fuel complex in summer is a likely explanation for this observation.

The analysis of our subset of data that had burn unit-specific bulk density showed there is a significant relationship between bulk density and spread rate. This relationship was similar to other shrubland fire behaviour studies [5] increasing bulk density (which in our case is also associated with increased fuel load) acted to reduce observed spread rate (Figure 6c). This result is likely due to the increased bulk density inhibiting the extent of heat transfer to the fuels ahead of the fire. Increased buoyancy created by the additional fuel loads consumed in the fuel complex could also lead to a reduction in heating ahead of the fire. It could also be that the high moisture content in the newly flushed deciduous shrubs slowed spread because of the additional heat required to evaporate the live moisture. However, we do not have the data resolution to examine this further since all shrub foliage (deciduous and evergreen), flowers and buds were sampled together. The LMC of deciduous species varies greatly through the growing season, reaching a peak of 200 to 300% soon after bud break, decreasing once seasonal growth has finished and reaching values as low as 60% when leaves are dying in the fall [27]. In contrast, evergreen shrubs have more complex patterns of seasonal moisture content that are similar to the foliar moisture content in conifers. In general, old needles reach their lowest moisture content when new needles are being formed. In Canadian fire behaviour practice, this period is termed the 'spring dip' and corresponds to foliar moisture content as low as 85% while 120% is the average during the rest of the year [14]. The NS-1 fuel complex is a mix of both.

Overall however the main difference in the fuel complex between the summer and the spring is the presence of newly flushed foliage on deciduous shrubs in the summer (creating higher bulk density and higher fuel load, and likely a higher live moisture content in the elevated fuel complex overall).

Figure 5a shows that the dead moisture content in the NS-1 fuel complex was consistently lower than indicated by the hFFMC. It might be expected that spread rate predictions directly from the FBP System would be similarly biased since these depend on the FFMC though the FWI System's ISI [14]. This bias in moisture content estimates has been observed in other open fuel complexes [15] and might explain why the unmodified FBP System spread rate models have not shown good predictive power in other shrub studies [28]. Figure 5b shows that EMC could be used as a reasonable surrogate estimate of moisture in the dead and cured elevated fine diameter component of fuel complex, which has been shown to be important for the spread in shrublands [5]. Our fire behaviour observations also suggest the expected decrease in spread rate with increasing moisture content in this layer although the relationship was not statistically significant. Figure 6b shows a shift in the relation between the two

seasons. Spring burns had similar spread rates as the summer burns, with higher dead fuel moisture, implying that under similar moisture contents spring fires would have exhibited increased spread rate over the summer burns. In our analysis here we simply looked at differences in a linear correlation between elevated dead moisture content and spread rate and not the exponential relationships between moisture and spread rate seen in other studies [5]. Our observations did not cover the range in data needed to examine the non-linear nature of this relationship, though it has been well-established elsewhere [5].

Our comparison of the ability of existing shrubland spread rate models to predict spread rates in the NS-1 fuel type did not yield any consistent or strongly statistically significant results overall; these comparisons were done within each season only however and therefore represented quite small sample sizes (n = 7 (spring), n = 6 (summer)). Visually, several of the models seemed to fit the variability in the data reasonably, particularly in the spring season (Figure 7). If we exclude one outlier from the spring dataset (burn unit 12 which burned over a very short duration and may have been influenced by strong gusts that were recorded) the correlations between the spring observations and the existing model forms tested, all improve to around r = 0.8 and are marginally significant around the *p* = 0.05 level (Table A5). Several of these existing models require specific inputs of bulk density or fuel complex height, which were only available on a unit-to-unit basis for the summer burns in 2017.

With the widespread knowledge and use of the FBP System [14] in fire operations in Canada, we felt it important to explore prediction results from that system. While the fuel complex differs, we felt the openness and exposure and overall fuel complex height of the NS-1 fuel type in our study should compare best to the O-1b fuel type in the FBP System, which represents open, exposed standing grass. While the default loading in the O-1b fuel type was about 1/10th that observed in the NS-1 fuel complex, recent studies in grassland with significantly heavier loadings [15] have found a modified O-1 FBP fuel type to work quite reasonably as an indicator of spread. The primary drivers of spread are wind speed and dead fuel moisture. As in that previous study [15], a modified version of the FBP System fuel type spread model, using an estimate of exposed moisture instead of the FFMC-based estimate of moisture, was found to provide improved prediction bias coupled with reasonable correlation with observed values.

The use of the O-1b fuel type also allows us to attempt to differentiate the spring and summer condition through the use of the curing function in the FBP System [14,29]. This function is used to reduce spread in a consistent fashion as the fuel complex becomes increasingly green, or in the case of grasslands as standing green grass senesces after the end of the growing season. In the FBP System, the effect of greenness (live vegetation) in the grass fuel complex is represented by an estimate of the amount of cured fuels in the fuel complex overall, a percentage cure value of 100% yielding maximum spread rate and representing no live vegetation component. In the case of the NS-1 fuel complex we are not drawing the analogue to cured fuels per se, but are using the curing function simply as a convenient indicator of the amount of greenness in the fuel complex and its potential impact on the spread. Our experimental results here, however, did not allow us to examine the change in spread rate with increasing greenness in the fuel complex. This suggested modifier is simply meant as a pragmatic way to use the existing FBP System model to achieve an appropriate adjustment for the increase live component in the fuel complex between the seasons. Our observations indicated that, overall, in the summer period spread rate was about 20% of what that predicted in the fully cured O-1b model, after accounting for wind and dead moisture content. A percent cure of 60% provides an approximation of this reduction using the FBP System's revised curing function [25] and is also readily applied using the common lookup tables used in the field by fire management personnel [30]. Spring spread rate in the fuel complex could be approximated using 100% cured state, again with the provision that we are using it as a measure of the state of the annual live vegetation in the fuel complex and not the overall state of the live fuel moisture in the entire fuel complex.

Using these initial guidelines, existing operational field guides [30] could be used to provide a reasonable first estimate of expected fire spread in the NS-1 type in spring or summer in the absence

of better estimates. As an aid to operational fire prediction in shrubland, we provide in Table A7, a quick lookup table to find the modified FFMC value that can be used in the field. Table A7 uses temperature and relative humidity to provide an effective (or modified) FFMC which can be used in the tables provided in the FBP System's "Red Book" field guide [30] to estimate an ISI and then a spread rate prediction. Due to the increased fuel loads, however, fire intensity in the fuel complex would be considerably higher than indicated by the O-1b fuel type. Our measurements in NS-1 indicate fuel consumed is roughly 7–10 times the FBP System standard amount assumed consumed in O-1b [30]. Scaling of fireline intensity with fuel consumption is a simple linear function [16] and therefore one would simply expect intensities to be 7–10 times higher for the same spread rate.

**Author Contributions:** Conceptualization, A.-C.P.; methodology, A.-C.P. and M.W.; formal analysis, A.-C.P. and M.W.; writing—original draft preparation, A.-C.P.; writing—review and editing, M.W.; supervision, M.W.; project administration, A.-C.P.; funding acquisition, A.-C.P. All authors have read and agreed to the published version of the manuscript.

**Funding:** This research was funded by Parks Canada, through the Conserve and Restore Program 2014–2019.

**Acknowledgments:** This project would not have been possible without significant in-kind contributions from the province of Nova Scotia (NSLF), who provided helicopter and fire crews during the burn operations. Dustin Oikle and Morgan Oikle, (NSLF), Tim Lynham and John Studens (Natural Resources Canada) also provided in kind support, helping with fuel sample collection and drying and Dan Thompson (Natural Resources Canada) processed the heat of combustion samples. Grant Pearce (Scion Rural Fire Research Group, New Zealand) reviewed results from after the first set up burns in 2014 and provided advice to improve the methodology and sampling effort for the 2017 set of burns. Matt Davies (Ohio State University) also provided some valuable reviews of an earlier report on the results of both experimental fires.

**Conflicts of Interest:** The authors declare no conflict of interest.

## Appendix A

**Table A1.** Summary of wind observations for the burning window at the Paquette Lake fires.

| Unit | Date | Start Time | End Time | 10 m WS (Top of Previous Hour) * | 10 m WS (Top of Following Hour) * | 10 m WS (Top of 2nd Previous Hour) * | Max Gust over Both Hours | 10 m WS During Burn (Via radio) ** | 10m WS Final |
|---|---|---|---|---|---|---|---|---|---|
| 7 | 06/10/2014 | 15:01:03 | 15:06:12 | 16 | 13 | | 31 | na | 16 |
| 12 | 06/10/2014 | 15:45:47 | 15:56:00 | 16 | 13 | | 31 | na | 13 |
| 17 | 06/10/2014 | 16:27:36 | 16:41:32 | 13 | 13 | | 20 | na | 13 |
| 8 | 06/10/2014 | 18:57:46 | 19:13:31 | 15 | 11 | 6 | 22 | na | 11 |
| 9 | 06/10/2014 | 19:35:00 | 20:18:00 | 11 | 6 | 9 | 12 | na | 9 |
| 32 | 06/12/2014 | 11:27:37 | 11:47:31 | 12 | 12 | | 24 | na | 12 |
| 27 | 06/12/2014 | 13:28:28 | 13:42:29 | 9 | 9 | | 10 | na | 9 |
| 22 | 06/12/2014 | 15:14:10 | 15:23:06 | 8 | 7 | | 8 | na | 8 |
| 13 | 07/14/2017 | 10:58:40 | 11:02:47 | 13 | 14 | | 22 | na | 14 *** |
| 14 | 07/14/2017 | 13:21:46 | 13:31:36 | 16 | 12 | | 28 | 14 | 14 |
| 24 | 07/14/2017 | 15:13:24 | 15:36:45 | 20 | 18 | | 29 | 20 | 20 |
| 25 | 07/14/2017 | 17:08:04 | 17:18:22 | 15 | 16 | | 28 | 15 | 15 |
| 19 | 07/14/2017 | 18:05:24 | 18:30:30 | 16 | 15 | | 27 | na | 16 |
| 23 | 07/15/2017 | 10:52:03 | 11:08:14 | 16 | 20 | | 20 | 20 | 20 |
| 18 | 07/15/2017 | 12:15:33 | 12:25:29 | 23 | 26 | | 42 | 20 | 20 |

* winds from the hourly station were only recorded for the 10 min at the top of each hour. Observations are reported for the observations over the entire duration of burning. ** wind speed recorded is a roving 10 min average of the 10 min previous to the log. *** FBAN recorded 5 km/h during burn unit 13.

*Appendix A.1. Live Moisture Content*

In order to estimate live moisture content for the site at the time of the burn, we added the fractional contribution of LWMC for each sample categories based on overall mean fuel load measurements. Since there was only one fuel load sample plot collected without shrub foliage on, which would have represented the site at the time of the 2014 burns, we use the overall average from the 35 sample plots

taken in 2015 but removed the shrub foliage load. This is not exact, since *Kalmia angustifolia* retains some foliage year-round but is the closest estimate we could use to estimate LWMC for the spring data set (Table A1).

**Table A2.** Spring 2014 burn Live Moisture content estimates based on fuel loading in the sampled categories.

| Spring 2014 | Load (kg/ m$^2$) | Load Proportion | Measured LMC (%) | Fractional Contribution LWMC (%) |
|---|---|---|---|---|
| Livefoliage | 0.3 | 27 | 183 (s.d. 31) | 49 |
| Live stems 0–1 cm | 0.8 | 73 | 101 (s.d. 5) | 73 |
| Total | 1.1 | 100 | | 122 |

**Table A3.** Summer 2017 burn Live Moisture content estimates based on fuel loading in the sampled categories. * sampled quantities have estimated standard deviation in parenthesis.

| Summer 2017 | Load (kg/m$^2$) | Load Proportion | Measured LMC (%) | Fractional Contribution LWMC (%) |
|---|---|---|---|---|
| Live shrub foliage | 0.1 (s.d. 0.06) | 11 | 207 (s.d. 88) | 23 |
| Live spruce foliage | 0.3 (s.d. 0.03) | 21 | 155 (s.d. 25) | 33 |
| Live stems 0–1 cm | 0.8 (s.d.0.19) | 68 | 104 (s.d. 7) | 71 |
| Total | 1.2 (s.d. 0.26) | 100 | | 127 |

*Appendix A.2. Within Plot Layout*

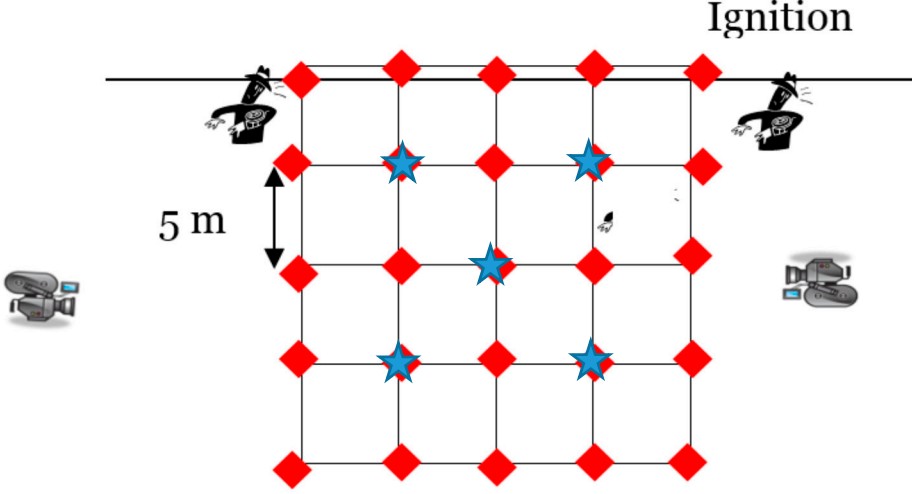

**Figure A1.** Plot layout indicating the location reference stakes defining the measurement grid and thermocouples layout (stars show thermocouple positions).

**Table A4.** Fuel consumption for Paquette Lake Summer 2017 burn units (n = 5/burn unit).

| Burn Unit | Pre-Load * (kg/m$^2$) | Post Load 0–1 cm (kg/m$^2$) | Estimated Fuel Consumption (kg/m$^2$) ** |
|:---:|:---:|:---:|:---:|
| 13 | 3.15 (s.d. 1.10) | 0.18 (s.d 0.08) | 2.96 (0.81) |
| 14 | 2.09 (s.d. 0.57) | 0.09 (s.d. 0.02) | 2.00 (0.57) |
| 18 | 3.05 (s.d. 1.06) | 0.14 (s.d. 0.06) | 2.91 (1.06) |
| 19 | 3.12 (s.d. 1.87) | 0.26 (s.d. 0.21) | 2.87 (1.88) |
| 23 | 3.19 (s.d. 1.48) | 0.12 (s.d. 0.09) | 3.07 (1.48) |
| 24 | 4.86 (s.d. 3.57) | 0.33 (s.d. 0.28) | 4.52 (3.58) |
| 25 | 2.77 (s.d.0.62) | 0.15 (s.d. 0.07) | 2.61 (0.52) |
| Mean | 3.17 (s.d. 0.84) | 0.18 (s.d. 0.09) | 2.99 (0.84) |

* pre-load includes litter, foliage and stems 0–1 cm. ** The standard deviation estimate on consumption is calculated as the square root of the sum of the estimated variance of the pre-burn and post-burn load.

**Table A5.** Comparison of various shrub fire behaviour models with the outlier point from Unit 12 removed. (This only influences the Springtime results).

| Model | Spring (n = 6) | | |
| | r (p) | Bias * | MAE * |
|:---:|:---:|:---:|:---:|
| Anderson et al. (2015) [5]: bulk density | 0.85 (0.03) | −43.0 | 3.0 |
| Anderson et al. (2015) [5]: height | 0.84 (0.04) | −2.7 | 2.7 |
| Catchpole et al. (1998) [18] | 0.84 (0.04) | 0.54 | 0.61 |
| Davies et al. (2009) [2] | 0.84 (0.04) | 1.7 | 1.7 |
| Fernandes et al. (2000) [3]: height | 0.85 (0.03) | −2.5 | 2.5 |
| Fernandes (2001) [4] | 0.83 (0.03) | −2.6 | 2.6 |
| US BEHAVE [13] | 0.76 (0.08) | 3.5 | 3.5 |
| FBP (O-1b) [14] | 0.97 (0.0009) | 5.1 | 5.5 |
| FBP (O-1b-modified) [14,15] ** | 0.86 (0.06) | 1.5 | 2.9 |

* Bias is average of Predicted–Observed. MAE is mean absolute error. ** Modification uses EMC (from the FFMC in the FWI System [10]) in place of moisture from the FFMC in the ISI calculation (see Appendix A Table A7). Summertime O-1b model is calculated using 60% of the fuel complex as being cured, springtime calculation uses 100%.

**Table A6.** Fuel loading per sample type for each summer burn unit.

| Burn Unit | Litter | Live Stems—0–1 | Live Stems—1–7 | Live Foliage—Coniferous | Live Foliage—Shrub | Dead Stems—0–1 | Dead Stems—1–7 | Total Load Total |
|---|---|---|---|---|---|---|---|---|
| 13 | 0.56 (0.36) | 1.07 (0.24) | 0.58 (0.62) | 0.34(0.19) | 0.14(0.04) | 0.27 (0.09) | 0.17(0.09) | 3.15 (1.1) |
| 14 | 0.89 (0.28) | 0.56 (0.23) | 0.14 (0.23) | 0.2 (0.12) | 0.09(0.04) | 0.09(0.04) | 0.11 (0.12) | 2.09 (0.57) |
| 18 | 1.01 (0.35) | 0.75(0.38) | 0.56(0.58) | 0.30 (0.30) | 0.10 (0.03) | 0.11 (0.08) | 0.21 (0.25) | 3.04 (1.06) |
| 19 | 1.10 (0.68) | 0.81(0.25) | 0.51 (0.71) | 0.25(0.18) | 0.19(0.10) | 0.12(0.07) | 0.14 (0.25) | 3.12 (1.87) |
| 23 | 0.81 (0.71) | 0.91 (0.30) | 0.37 (0.30) | 0.24 (0.11) | 0.13(0.09) | 0.25(0.11) | 0.47(0.45) | 3.19 (1.48) |
| 24 | 2.05 (1.71) | 1.01(0.20) | 0.48 (0.44) | 0.33(0.14) | 0.13 (0.04) | 0.33 (0.26) | 0.51 (0.66) | 4.86 (3.57) |
| 25 | 0.98 (0.20) | 0.82 (0.30) | 0.34(0.26) | 0.20(0.11) | 0.14(0.04) | 0.15(0.06) | 0.15 (0.24) | 2.77 (0.62) |
| 2014 mean | | | | | | | | 3.5 ± 1.17 |
| 2017 mean | | | | | | | | 3.17 ± 0.83 |

**Table A7.** Modified FFMC value for the NS-1 shrubland fuel complex. The method uses the Equilibrium Moisture Content model from the FWI System to estimate moisture content using temperature and relative humidity observations.

| RH (%) | Temperature (Degrees C) | | | | | |
|---|---|---|---|---|---|---|
| | 5 | 10 | 15 | 20 | 25 | 30 |
| 10 | 95 | 95 | 96 | 97 | 97 | 98 |
| 20 | 92 | 92 | 93 | 94 | 95 | 96 |
| 30 | 89 | 90 | 91 | 92 | 93 | 94 |
| 40 | 87 | 88 | 89 | 90 | 91 | 92 |
| 50 | 86 | 86 | 87 | 88 | 89 | 90 |
| 60 | 84 | 85 | 86 | 86 | 87 | 88 |
| 70 | 82 | 83 | 84 | 85 | 85 | 86 |
| 80 | 80 | 81 | 82 | 82 | 83 | 84 |
| 90 | 77 | 78 | 78 | 79 | 80 | 81 |
| 100 | 71 | 72 | 72 | 73 | 74 | 74 |

Values in the table cells correspond to an effective hFFMC (or FFMC) value. Temperature and humidity are used to estimate an EMC value from the FWI System's FFMC model calculation (average of drying and wetting). Then that EMC value (which is moisture content in %) is converted to an equivalent "Code" value using the FFMC's FF-scale [10].

This new 'effective FFMC' is then used along with wind to estimate a new ISI value (using either the equations within the FBP System [14] or the FBP System "Red Book" Field Guide-Table 6.3 [30]).

The spread rate can then be estimated using that new ISI and the equations for Spread rate in O-1b [14] or the FBP System "Red Book" Field Guide-Table 9.24 [30].

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
