# Peer review of "Fire Behaviour Observation in Shrublands in Nova Scotia, Canada and Assessment of Aids to Operational Fire Behaviour Prediction"

_fire, doi:10.3390/fire3030034_

Round 1
Reviewer 1 Report
I'm excited to see this research submitted for publication. This paper reports important results from a unique wildland fuel type. The findings have significance for both expanding the scientific evidence base on shrubland fire behaviour, and for operational forecasting of potential fire danger. The authors have completed a detailed set of measurements of multiple fire behaviour parameters and important potential controlling variables using a well-designed experiment. The research was impacted by a not-unusual array of logistical difficulties but the authors have sought to circumnavigate them where possible. Importantly the authors are up-front about these issues and their conclusions contain appropriate caveats.
However, though the data and results reported here have significant scientific value there are a number of areas where improvements to the manuscript can be made. These are necessary to ensure readers know how the data was collected and analysed. The following general recommendations can be made:
1) A thorough review of the structure and organization of the paper is necessary to ensure information is reported in the appropriate section. For instance there are a number of places where important methodological details are reported in the results, and some parts of the results would seem to be better placed in the discussion.
2) I generally appreciate brevity in Introductions but this section does need to contextualize the manuscript within previous research on shrub fire behaviour, fire danger rating and fuel moisture dynamics - all topics explicitly studied in the paper. The introduction defines to nature of the specific problem well but it would be helpful to provide additional detail and background on international work to understand shrub fire behaviour. This should consider the theoretical and practical challenges to shrub fire modeling and the ecological/environmental importance of understanding fire behaviour and effects in these and similar systems.
3) The Introduction provides an overall aim for the paper but I would like to see specific objectives defined which thereafter guide the structure of the results and the data analyses presented.
4) Additional detail and re-organization is required within the Methods section. A number of specific issues are flagged in the attached annotated manuscript. More significant issues for attention include:
a) More detail on the study system - The map (Figure 1) is great but the nature of the study system needs to be better explained for an international audience. A little more information on the biophysical characteristics of the region would be helpful - this could include information on local climatic conditions, soil characteristics and vegetation composition. Some information on the history of fire in the region would also be useful.
b) Ensure consistency in terminology used to describe elements of the sampling design (e.g. units, plots, quadrats etc.)
c) Currently substantial methodological information is reported in the results section (flagged in annotated version). This pertains to, for example, sampling of fuel moisture and fuel consumption, and methods used to model fire behaviour.
d) Information on data analysis procedures, including statistical packages used, must be provided. It should be clear how analyses align with the paper's specific objectives. A proportion of the information presented (e.g. on fuel moisture dynamics and fuel structure characterization seems of secondary importance to the focus suggested by the title of describing and modeling fire behaviour.
e) I have a number of specific methodological queries including:
i) Why was 10m wind speed used to model fire spread? I believe approximate mid-flame height would be the norm
ii) How was Equilibrium Moisture Content estimated?
iii) How was Byram's fireline intensity (HwR) estimated for plots lacking specific fuel consumption estimates?
iv) The authors compare their RoS data to a comprehensive array of existing spread models which is great. I'm curious why they did not seek to develop their own empirical model?
v) How were estimates of fuel load made for plots lacking fuel quadrats (i.e. the 2014/spring burns)?
vi) The authors chose to examine associations between RoS and FBP predictions but did not examine correlations between RoS and ISI or intensity and FWI. What was the reason for this?
5) A number of clarifications need to be made in the presentation of the results.
a) Ensure all means are accompanied by an estimate of variability - this is done in some cases and not in others. On some occasions the standard error is reported, in others the standard deviation. I think the latter is most informative
b) In the tabulation of the results (e.g. Table 1) and relevant figures the authors need to make it clear for which burns: i) site level fuel structure/load estimates are used; ii) off-site wind speed estimates are used. The latter has a major consequence for subsequent analyses but it's never abundantly clear where accurate on site data is available. Does this, for example, explain the poor prediction of RoS in the 2017/summer fires?
c) In situations where fitted lines are presented the methods behind these, and the results of the analyses should be presented (e.g. Figure 4). In Figure 4 specifically it's apparent that different relationships may exist between dFMC and hFFMC for different fuel types. What is the justification for fitting a single line? Did the authors examine fuel type x hFFMC interactions?
d) Many of the correlations reported between fire behaviour and potential explanatory factors are non-significant but the results are still presented as if meaningful relationships exist (e.g. lines 307-313). What is the justification for this?
e) Some of the figures require larger text and clearer symbology to make them easy to interpret. I would recommend preparing such figures in a package such as R or SigmaPlot to ensure they are of high quality.
6) The discussion provides a good interpretation of some of the results but I would like to see additional explanation/focus on three issues:
a) The reason for the poor RoS model performance for summer/2017 fires. Why do most empirical models, but not the FBP, system do so poorly for these burns?
b) The overall strong performance of the FBP system contrasts with previous studies from, for example New Zealand and the UK, which has found difficulty using this system for shrub fuels. What did you do differently and what are the lessons for shrub fire danger rating more widely (given the CFWIS is adopted widely)?
c) It would be beneficial to make some quantitative comparison of the actual spread rates and intensities seen here and those encountered in other shrublands. The results could, therefore, be more strongly contextualized against previous shrub fire behaviour research.
I hope these suggestions are helpful in improving this important and potentially impactful paper. I am at the authors' disposal should they wish me to clarify any of my comments or if they believe I can be of further help.

Author Response
Thank you for your feedback. Your comments have been addressed and a response to each is found in attachment. A thorough restructuration of the manuscript has been done to ensure a better separation of methods, results and discussion, with clearer objectives and a longer introduction for better context.

Reviewer 2 Report
see comments attached

Author Response
Thank you for your feedback. Your specific comments have been addressed and a response to each is found in attachment. The revised manuscript has been restructured to improve the separation between methods, results and discussion.

Reviewer 3 Report
The study reports fire behaviour data acquired in shrubland burning in Canada’s Cape Breton, as well as the associated fuel and weather conditions. As the Canadian FWI System is used more and more outside Canada this study could expand its ability to portray fire danger and behaviour in woody live fuel complexes other than conifer crown fires. However, the study scope is short, given the number of successful experiments (n=13), size of the plots (only 20-m wide) and problems in recording the “real” wind speed driving fire spread. Still, I think it warrants publication, given it provides fire behaviour data from outdoors experiments in a fuel complex for which no data/knowledge are available.
Below I offer specific comments, which are mostly related to the need of clarifying methods and reorganize or expand sections of the text.
Introduction: I am not an adept of long Introductions, but in this manuscript the extent is definitely too short. Please elaborate more on shrubland fire behaviour and on the context and needs.
L37. Reference for this?
L81. Time and temperature of oven drying?
L98. Please provide latin names.
L114. What was the temperature criteria for determining time of arrival?
L125. Correct “ :”
L201. Substantial parts of section 3.1 and also 3.2 and 3.3 are methods rather than results and so should be displaced.
L269. Is this EMC the output from Van Wagner (1972) model?
L276. Avoid repeating values included in Table 1, and mention Table 1 in the text.
L327-346. Almost nothing is said about the results of the comparisons.
L336. Typo.
L344. Cite Wotton (2009) as the modified version.
L353. Described what is r (p).
L357. I don’t think it makes sense to include heat of combustion here. Instead, move it to before fire behaviour – next to fuel load.
L361. Unclear what “(-2.4 MJ - 100%MC)” means.
L362. Fire intensity.
Author Response
Thanks for the feedback. Your specific comments have been addressed and are found in attachment. The revised version of the manuscript has been restructured to have a better separation of methods, results and discussion and flow better.
